# Clinical outcomes of chikungunya: A systematic literature review and meta-analysis

**Kris Rama**[1]*, **Adrianne M. de Roo**[2,3], **Timon Louwsma**[1,3], **Hinko S. Hofstra**[1], **Gabriel S. Gurgel do Amaral**[1], **Gerard T. Vondeling**[2], **Maarten J. Postma**[3,4,5,6], **Roel D. Freriks**[1,3,7]

**1** Asc Academics B.V., Groningen, Netherlands, **2** Valneva Austria GmbH, Vienna, Austria, **3** Department of Health Sciences, University Medical Center Groningen, Groningen, Netherlands, **4** Department of Economics, Econometrics & Finance, University of Groningen, Faculty of Economics & Business, Groningen, The Netherlands, **5** Center of Excellence for Pharmaceutical Care Innovation, Universitas Padjadjaran, Bandung, Indonesia, **6** Division of Pharmacology and Therapy, Faculty of Medicine Universitas Airlangga, Surabaya, Indonesia, **7** Department of Health Technology and Services Research, University of Twente, Enschede, The Netherlands

* krisrama94@gmail.com

**Data Availability Statement:** All relevant data are within the manuscript and its Supporting information files.

## Abstract

### Background

Chikungunya is a viral disease caused by a mosquito-borne alphavirus. The acute phase of the disease includes symptoms such as fever and arthralgia and lasts 7–10 days. However, debilitating symptoms can persist for months or years. Despite the substantial impact of this disease, a comprehensive assessment of its clinical picture is currently lacking.

### Methods

We conducted a systematic literature review on the clinical manifestations of chikungunya, their prevalence and duration, and related hospitalization. Embase and MEDLINE were searched with no time restrictions. Subsequently, meta-analyses were conducted to quantify pooled estimates on clinical outcomes, the symptomatic rate, the mortality rate, and the hospitalization rate. The pooling of effects was conducted using the inverse-variance weighting methods and generalized linear mixed effects models, with measures of heterogeneity reported.

### Results

The systematic literature review identified 316 articles. Out of the 28 outcomes of interest, we were able to conduct 11 meta-analyses. The most prevalent symptoms during the acute phase included arthralgia in 90% of cases (95% CI: 83–94%), and fever in 88% of cases (95% CI: 85–90%). Upon employing broader inclusion criteria, the overall symptomatic rate was 75% (95% CI: 63–84%), the chronicity rate was 44% (95% CI: 31–57%), and the mortality rate was 0.3% (95% CI: 0.1–0.7%). The heterogeneity between subpopulations was more than 92% for most outcomes. We were not able to estimate all predefined outcomes, highlighting the existing data gap.

**Funding:** This paper was funded by Valneva Austria GmbH. AMR and GTV are Valneva employees. The funder had no role in study design, data collection and analysis, decision to publish, or preparation of the manuscript.

**Competing interests:** I have read the journal's policy and the authors of this manuscript have the following competing interests: KR, TL, HSH, and GSG are employees of Asc Academics. Asc Academics has received consultancy fees for this project from Valneva Austria GmbH. AMR and GTV are Valneva employees and own stock options of Valneva. MJP reports grants and honoraria from various pharmaceutical companies, including those developing, producing, and marketing vaccines. He holds stocks in Health-Ecore (Zeist, Netherlands) and PAG BV (Groningen, Netherlands), and advises ASC Academics (Groningen, Netherlands). These competing interest will not alter adherence to PLOS policies on sharing data and materials.

## Conclusion

Chikungunya is an emerging public health concern. Consequently, a thorough understanding of the clinical burden of this disease is necessary. Our study highlighted the substantial clinical burden of chikungunya in the acute phase and a potentially long-lasting chronic phase. Understanding this enables health authorities and healthcare professionals to effectively recognize and address the associated symptoms and raise awareness in society.

## Author summary

Chikungunya disease is an emerging public health concern. The disease is transmitted by mosquitoes and characterized by arthralgia and fever in the acute phase, lasting 7–10 days. Additionally, some individuals experience chronic symptoms such as arthralgia and tiredness that can last from months to years. Chikungunya is mainly present in the Americas and Asian countries, but the mosquitoes transmitting the disease are spreading to other regions due to climate change, amongst others. This increased disease threat highlights the importance of understanding chikungunya symptoms. However, there are currently no precise estimates on the prevalence of chikungunya symptoms. Therefore, we analysed the available literature on the clinical manifestations of chikungunya. We found that 75% of infected people develop symptoms, primarily characterized by arthralgia in 90% and fever in 88% of cases. Chronic symptoms affect 44% of symptomatic people, and 0.3% of patients with chikungunya die. Unfortunately, we were not able to estimate all predefined outcomes of interest because we did not find enough studies publishing on some of these, demonstrating that there is still much unknown around the clinical manifestations of chikungunya. However, the results can help healthcare workers early identifying chikungunya and raise awareness of this debilitating disease.

## Introduction

Chikungunya is a viral disease caused by a mosquito-borne alphavirus, the chikungunya virus (CHIKV) [1]. The infection is characterized by an acute phase with symptoms including fever, arthralgia, and myalgia. While most infected individuals fully recover after the acute phase of the disease, between 30–40% of patients develop persistent symptoms, such as chronic arthritis, fatigue, stiffness, depression, and sleep and neurological disorders, which can last from months to several years [2, 3]. Long-term effects lead to significant limitations in daily activities and reduce the patients' overall quality of life [4–6]. Nevertheless, despite the negative impact of the disease on the quality of life, the awareness and societal understanding of chikungunya remain limited, even among the afflicted populations and healthcare workers [7]. Chikungunya has been identified as a public health threat based on several records of CHIKV outbreaks worldwide, with a risk of exacerbation in the future due to the global spread of CHIKV [8]. The distribution of the CHIKV vectors (*Aedes aegypti* and *Aedes albopictus*) is one of the main factors contributing to the disease's dissemination. This expansion is attributed to globalization and climate change, allowing the *Aedes* mosquitos to survive in areas previously considered unsuitable [9, 10]. Prevention against the disease consists predominantly of mosquito population control [11]. Recently, the FDA approved the first chikungunya vaccine, presenting a new tool to fight the disease and potentially alleviate the associated economic and

health burdens [12]. Despite the increasing interest in CHIKV and the recent announcement of a vaccine, uncertainties persist regarding the clinical burden of chikungunya. Although multiple studies have explored one or more health outcomes associated with chikungunya [3, 13, 14], to the best of our knowledge, no extensive meta-analysis was performed to quantify pooled estimates on the clinical presentation of chikungunya. To address this gap, we conducted a comprehensive systematic literature review (SLR) on the clinical manifestations of chikungunya, and proceeded with a robust yet flexible meta-analysis. This approach allowed us to provide estimates on a broad spectrum of endpoints on the health outcomes of chikungunya. We paid particular attention to the symptomatic, mortality, and chronicity rates for a comprehensive understanding of the disease in both acute and chronic phases. Our study aims to contribute valuable insights into the overall clinical outcomes of chikungunya. This, in turn, can inform public health intervention strategies and enhance global surveillance by enabling earlier detection of outbreaks.

## Methods

### Literature search and study selection

The SLR adhered to the Preferred Reporting Items for Systematic Reviews and Meta-Analyses 2020 (PRISMA 2020) guidelines, with searches conducted on MEDLINE In-Process via PubMed.com, and Embase via Embase.com without time limits. Grey literature searches were performed for the years 2019–2023 to capture data that may not have yet been included in the databases. The search string included terms related to chikungunya and study design. Eligibility criteria were developed using a Population, Intervention, Comparator, Outcomes, Study (PICOS) framework. The inclusion criteria focused on the clinical manifestations of chikungunya, their prevalence and duration, and related hospitalization, and excluded in vitro/preclinical studies, reviews, and non-English articles. Specifics can be found in S1 Text.

### Screening and data extraction

All retrieved articles were deduplicated and titles and abstracts were screened against the PICOS criteria using Rayyan. From the selected articles, full texts were examined for eligibility, followed by detailed data extraction organized by study design, patient characteristics, and outcomes of interest. The whole screening process was conducted by two independent reviewers (GG, HH), resolving conflicts through consensus. An exhaustive feasibility assessment ensured the inclusion of studies with explicit criteria and comparable reporting methods, reducing heterogeneity and potential outlier influence. The risk of bias was determined using a modified Downs and Black checklist [15] and NIH quality assessment tool for observational studies [16], see S2 Text. Discrepancies were resolved by consensus. No protocol for this systematic review and meta-analysis was registered.

### Population and data analysis

The meta-analysis was performed using the meta package of the R statistical software to create a pooled estimate of the most important clinical outcomes of chikungunya. The outcomes of interest were the overall symptomatic, chronicity, and mortality rates, the underreporting factor, the duration of the acute and chronic phase, the hospitalization and outpatient rate (acute and chronic), the mortality rate per region, and the rate and duration of the following symptoms: arthralgia, arthritis, fatigue, fever, headache, joint swelling, myalgia, nausea, rash, and vomiting. The distinction between arthralgia and arthritis was made based on the definition used in the original study.

Both fixed-effects and random-effects models with logit transformation were estimated, where a random-effects model was chosen in case of high heterogeneity. Fixed-effects meta-analyses employed inverse-variance weighting, while random-effects accounted for between-study heterogeneity and are better suited to account for the larger variations in outcomes reported. Heterogeneity was assessed using Cochran's $Q$, $I^2$, $H^2$ statistics, and $\tau^2$ estimation. Outlier analyses employed the leave-one-out method, Baujat plots, and statistical distance measures. All results were visually represented using forest plots, providing a clear and concise graphical representation of the individual study findings and the overall meta-analysis result.

Our study utilized subpopulations—subsets of the original populations defined by particular demographic and clinical features. These features correspond to the data reported in the studies we analyzed and the segmentation into subpopulations was based on the inclusion or exclusion criteria set forth in the original research papers. This approach allowed us to perform a more granular analysis. The clinical outcomes of interest were analyzed for a target population to ensure comparability among included studies, which excluded children under 15, individuals with comorbidities or concurrent infections, and pregnant women. Additionally, we excluded unconfirmed CHIKV cases and studies involving chronic patients reporting on the acute phase due to recall bias. Lastly, retrospective studies focusing on mortality were excluded as they exhibited evidence of selection bias. Meta-analyses were performed when an endpoint was reported at least five times for a given subpopulation.

A preliminary search indicated that data on chronicity, mortality, and symptomatic cases was predominantly reported for a more general population, including individuals under the age of 15 and chronic patients. Therefore, we decided to apply less strict criteria on the studies reporting these outcomes, allowing us to estimate these endpoints. Additionally, to detail the development of chronic symptoms, we estimated the chronic rate at various points from disease onset by dividing studies reporting on chronicity rates following a CHIKV infection into subgroups based on time intervals (three, six, and 12 months). The inclusion criteria for each subgroup were to fall within the time windows created by consecutive intervals (e.g., 90–180 days for three months). We excluded studies extending beyond 24 months to avoid a selection bias, as these already focused on patients with pre-existing chronic conditions.

For the mortality rates, we separated the groups that reported outcomes for high-risk populations from those dealing with the general population with lower risk. This stratification allowed us to account for potential confounding variables. Older age and comorbidities have been identified to increase the risk for mortality [2, 17]. Therefore, we classify as high-risk of mortality the groups with a minimum age over 65 (or median above 70 when missing), and previous conditions that induced prior intensive care exceeding 24 hours.

To estimate the overall symptomatic rate, we included studies that explicitly reported symptomatic rates based on one or more of the symptoms commonly associated with the disease. Symptomatic patients were often an implicit inclusion criteria, or a precondition for laboratory testing, making most of the studies reporting on the symptomatic rate unusable. We excluded the studies that had a 100% symptomatic rate to prevent selection bias, as including those would lead to a skewed perspective due to symptoms being part of their inclusion criteria.

## Results

### Literature search

The SLR was conducted on 4 July 2023 and yielded 16,308 hits. After removing 6,285 duplicates, 10,023 studies were screened by titles and abstracts. From these, a total of 316 articles were deemed suitable for inclusion. The process of the SLR is detailed in Fig 1, which illustrates

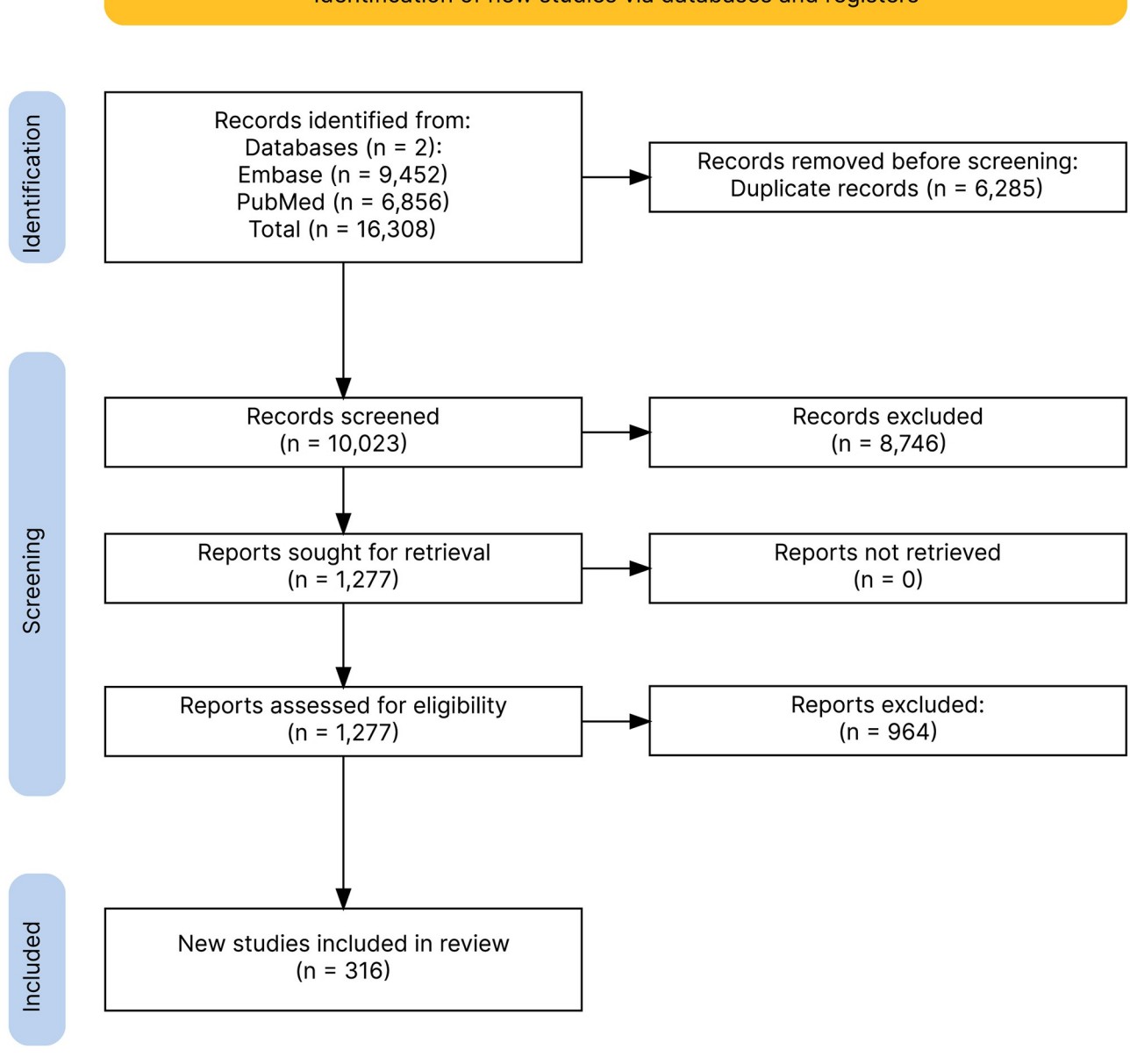

**Fig 1. PRISMA diagram of the included studies.**

the PRISMA diagram of the included studies. The complete PRISMA checklist is provided in S3 Text. The quality assessment of included studies can be found in S1 Table.

## Study characteristics

The categorization of study designs in the included articles was made with careful consideration, taking into account the diversity in how these studies defined their methodologies. The judgment used in categorizing these studies was guided by the definitions provided within the paper itself. When a study described its design in a way that matched more than one

predefined category, the predominant one was chosen. This approach aimed to respect the original terminology used by the study authors while also creating a coherent framework for analysis.

Of the 316 articles included, 231 studies were observational, 11 were experimental, and for 74 studies this was not reported. Of the observational studies, 106 were cross-sectional, 35 were cohort, 29 were longitudinal, 25 were retrospective, 23 were prospective, and 13 were case-control or case-series studies. Of the experimental studies, there were 6 trials from phase I to III with double and single-blind designs. Goals ranged from assessing treatments like chloroquine and vaccines' effectiveness to exploring seroprevalence and chronic CHIKV effects. Two trials investigated new mRNA treatment mechanisms. The focus was solely on CHIKV, not on coinfections.

The study location varied: Southern Asia was the most represented with 78 articles, followed by South America, with 67. There were 41 articles from The Caribbean region, 41 from Eastern Africa, 28 from South-Eastern Asia, 14 from Central America. Eight, six, five, four, and three articles were from Western Europe, Northern America, Southern Europe, Middle Africa, and Western Africa, respectively. Two or one articles were from Eastern Asia, Micronesia, Northern Europe, or Southern Africa. A total of 193 studies reported mean or median age. Data on co-infection with Zika and/or dengue were reported in 11 studies. An overview of the study characteristics, including details on the experimental studies, can be found in S2 Table.

The most commonly reported symptom was fever, reported in 57.9% of the studies (N = 183), followed by rash in 54.1% (N = 171), headache in 51.3% (N = 162), and arthralgia in 47.8% (N = 151). Most studies reported high rates (70% to 100%) of fever. Among the 151 studies reporting arthralgia rates, the symptom prevalence ranged from 1% to 100%, as studies presented heterogeneous settings, including, for example, recovered patients, patients in the acute phase, or chronic patients. Duration of symptoms was reported in 22 studies. Taking all symptoms into account, the mean duration of symptoms ranged from two days (fever) to 342 days (arthralgia). It is important to note that the studies presented heterogeneous groups of patients when reporting on the duration of symptoms, which could explain the wide range reported in literature. The hospitalization rate was reported by 53 studies. The hospitalization rate varied between 0%, reported by five different studies [18–22], and 93% in a study by Reller and colleagues [23].

The development of chronic disease after CHIKV infection was reported in 68 studies. Most studies defined chronic CHIKV infection as the presence of symptoms three months after the infection. Arthralgia was reported as a chronic symptom in 67 studies, joint swelling was reported in 11 studies, myalgia was reported in eight studies, stiffness, especially in the morning, was reported in six studies, and arthritis was reported in four studies. The percentage of patients developing chronic disease ranged from 16% in a study conducted during an outbreak in Chennai, India [24] to 100% in two other studies [25, 26]. Fifty of the included studies reported data on mortality, of which 22 reported no deaths in the study population. The highest reported mortality rate was 36.67%, or 36,670 per 100,000 population, reported by Gupta and colleagues. This study population consisted of chikungunya patients who had been admitted to the intensive care [27].

## Meta-analyses feasibility and results

From the 316 articles retrieved from the SLR, we extracted 756 distinct subpopulations. Each subpopulation corresponds to a group defined by a unique set of inclusion and exclusion criteria as per the definitions provided in each original study. Out of the 756 subpopulations, 335 were used for the analysis of the general population, while 52 where used for the target

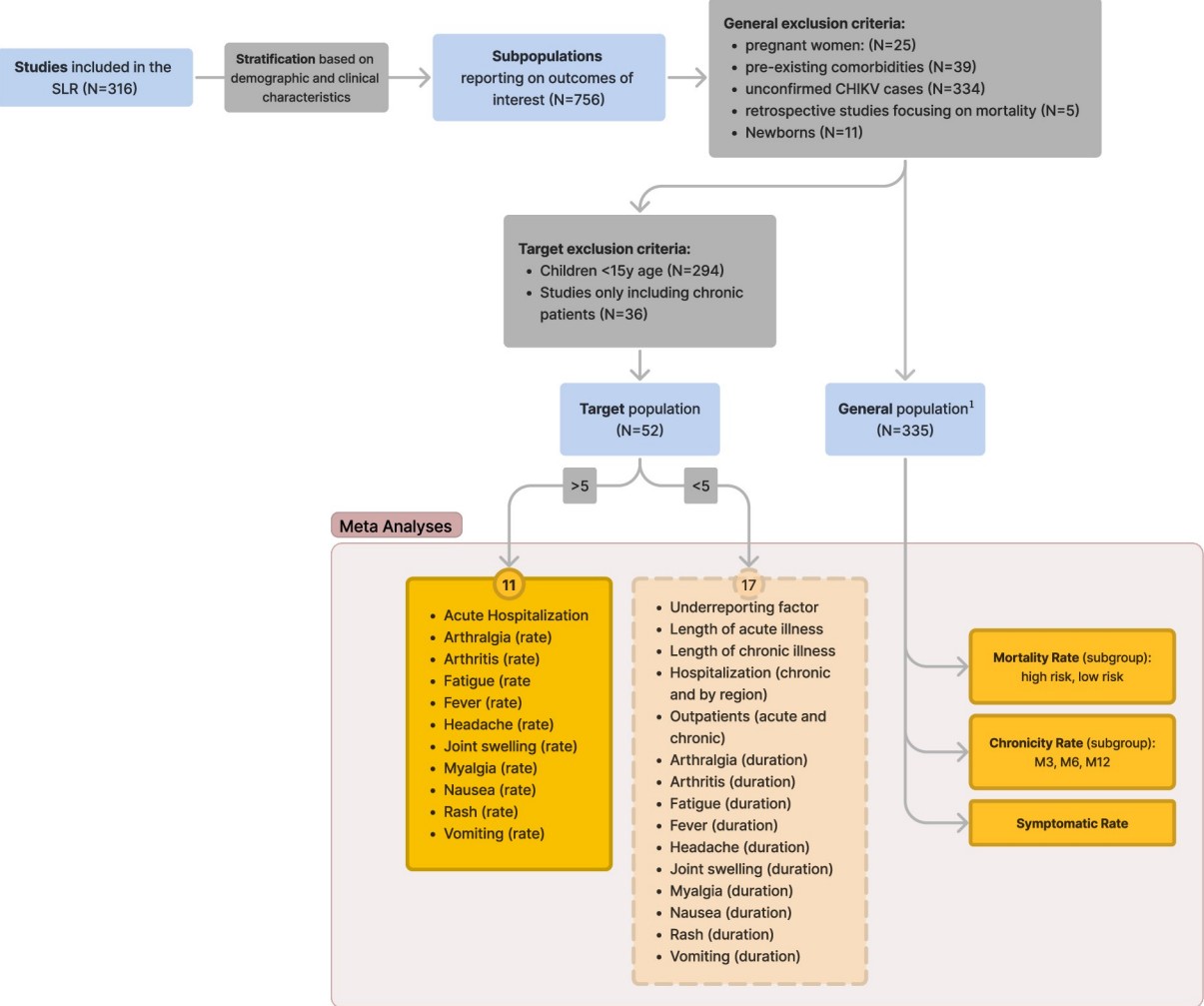

¹The number of studies providing data on mortality rate, chronicity rate, and symptomatic rate was limited. Therefore, we analysed these for the general population.

**Fig 2. Flowchart of the meta-analysis process and number of estimated endpoints.**

population. From the 28 selected clinical outcomes, we were able to conduct 11 meta-analyses for the target population, see Fig 2. The number of studies and subpopulations available for each endpoint is shown in Table 1. The forest plots from the individual meta-analyses can be found in S1 Fig, and the outlier analysis for each endpoint with the Baujat plot is presented in S2 Fig. No studies or subpopulations were excluded based on outlier analyses. Below, we present the 11 estimates from the meta-analyses on the target population, followed by the results of the analysis on mortality, chronicity, and overall symptomatic rates in the general population.

**Chikungunya symptoms estimates in the target population.** The prevalence of arthralgia in symptomatic adults with confirmed chikungunya was estimated at 89.7%, while arthritis was less frequent at 17.6%. Fatigue was observed in 56% of patients, fever in 87.8%, and headache affected 49.5% of the population. Joint swelling was reported in 50% of patients, myalgia in 62.9%, nausea in 34.7%, rash in 44.3%, and vomiting in 17.1%. The hospitalization rate during the acute phase of chikungunya was reported by nine subpopulations and estimated at 17%. All results are presented in Table 1, showing the pooled effect estimate for each symptom, reflecting the average rate of occurrence in the studied populations within specified confidence

**Table 1. Meta-analysis of the different clinical outcomes of chikungunya disease.** Presented are the number of studies and number of subpopulations reporting on the specific outcomes, the pooled estimates, confidence intervals and $I^2$ of the estimated endpoints. CI: confidence interval. $I^2$: I-squared statistic of heterogeneity.

| | Endpoint | Studies | Subpop | Total | Pooled | 95%CI | $I^2$ |
|---|---|---|---|---|---|---|---|
| **Target Population** | **Symptoms** | | | | | | |
| | Arthralgia | 22 | 30 | 3714 | .897 | .827–.940 | 93% |
| | Arthritis | 9 | 14 | 1622 | .175 | .089–.315 | 94% |
| | Fatigue | 7 | 10 | 1291 | .560 | .389–.718 | 95% |
| | Fever | 17 | 24 | 3490 | .878 | .849–.902 | 75% |
| | Headache | 18 | 25 | 2423 | .495 | .375–.616 | 95% |
| | Joint Swelling | 7 | 9 | 632 | .500 | .410–.589 | 85% |
| | Myalgia | 17 | 19 | 2588 | .629 | .483–.754 | 97% |
| | Nausea | 8 | 10 | 968 | .347 | .156–.604 | 96% |
| | Rash | 21 | 28 | 2990 | .443 | .350–.539 | 92% |
| | Vomiting | 11 | 17 | 1446 | .171 | .113–.249 | 88% |
| | **Hospitalization** | | | | | | |
| | Acute Phase | 9 | 9 | 1700 | .170 | .036–.528 | 97% |
| **General Population** | **Chronic rate** | | | | | | |
| | Month 3 | 17 | 17 | 2805 | .438 | .313–.572 | 97% |
| | Month 6 | 10 | 10 | 1899 | .343 | .248–.454 | 97% |
| | Month 12 | 11 | 11 | 2366 | .318 | .215–.443 | 96% |
| | **Mortality rate** | | | | | | |
| | High Risk | 5 | 7 | 325206 | .1534 | .0713–.2994 | 97% |
| | Low Risk | 30 | 37 | 2518 | .0032 | .0014–.0074 | 87% |
| | **Symptomatic rate** | | | | | | |
| | Overall | 8 | 8 | 1217 | .749 | .630–.840 | 91% |

intervals. Each symptom analysis showed substantial heterogeneity between subpopulations, indicated by high $I^2$ statistics.

**Chronicity, mortality, and overall symptomatic rate.** The meta-analysis for chronicity rate showed declining rates over time: 43.89% at three months, 34.39% at six months, and 31.87% at twelve months, see Fig 3. Notably, persistent high heterogeneity was observed across subgroups ($I^2$ between 96–97%). Mortality rates were estimated at 0.32% (320 per 100,000 population), for normal-risk populations and 15.34% (15,340 per 100,000 population) for high-risk populations, see Fig 4. The latter displayed higher heterogeneity ($I^2$ = 97%) compared to the normal risk ($I^2$ = 87%). The meta-analysis estimates that 74.9% of the general population with CHIKV infection were symptomatic, with a 95% confidence interval from 63% to 84%, see Fig 5. A total of eight studies with corresponding eight subgroups were included in this analysis. $I^2$ statistics showed a heterogeneity of 91%. Results of the outlier and influential cases analysis can be found in S2 Fig.

## Discussion

Chikungunya poses an emerging global health threat; however, uncertainties around the health burden of this infectious disease persist. This SLR and meta-analyses aim to consolidate existing research on the clinical manifestations of chikungunya. The objective of this study was to provide accurate estimates on the symptomatology of this disease, with a specific focus on the chronicity, mortality, and overall symptomatic rates. Overall, our findings emphasize the substantial disease burden associated with a CHIKV infection.

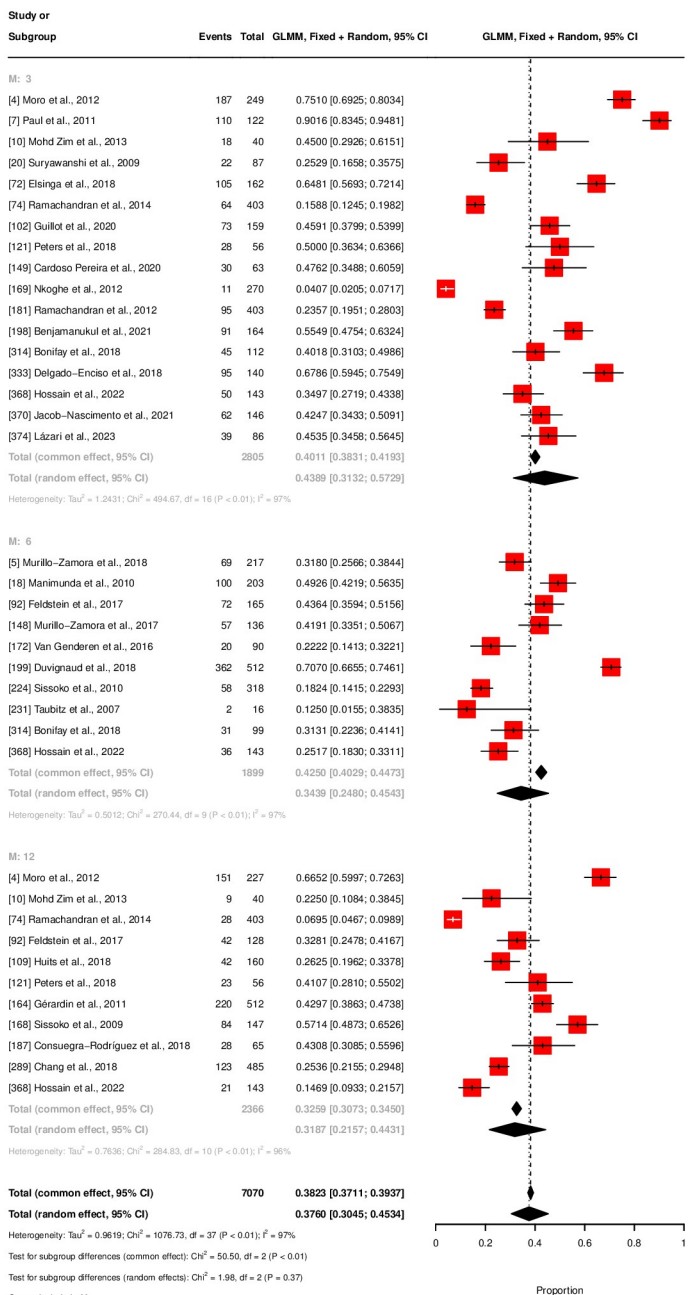

**Fig 3. Forest plot for chronicity rate.**

Arthralgia, fever, and myalgia were the most prevalent symptoms, affecting 89.7%, 87.8%, and 62.9% of symptomatic adults, respectively. These symptoms are also described in previous literature as most common for chikungunya [17, 28]. It's important to note that these symptoms were often implicitly used when initially detecting suspected cases. Although we removed all explicit inclusion criteria, these estimates are likely affected by selection bias. The hospitalization rate of 17% underscores the challenges for healthcare systems during outbreaks. The disease burden related to these symptoms makes chikungunya a significant burden for local healthcare systems, highly influencing the quality of life of infected individuals [6].

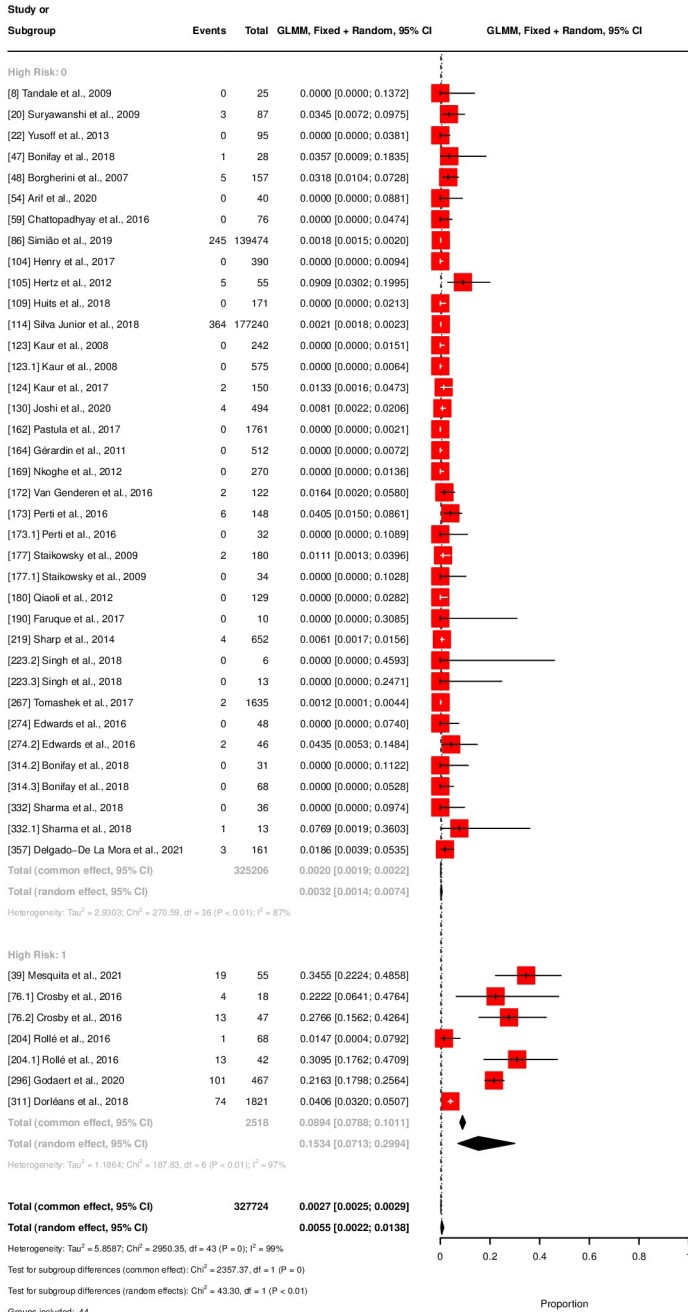

**Fig 4. Forest plot for mortality rate.**

The number of studies that provided data on mortality, chronicity, and overall symptomatic rate was limited for the target adult population. Thus, we decided to use less restrictive population criteria for these specific outcomes. Within this broader general population, we found a 0.32% (320 per 100,000 population) mortality rate in the low-risk group. This is slightly higher than the common reported case-fatality rate of 0.1% (100 per 100,000 population), although reports on mortality rated may vary [2, 6]. To our knowledge, no previous meta-analysis on mortality rates has been performed. Therefore, we argue that 0.32% (320 per 100,000

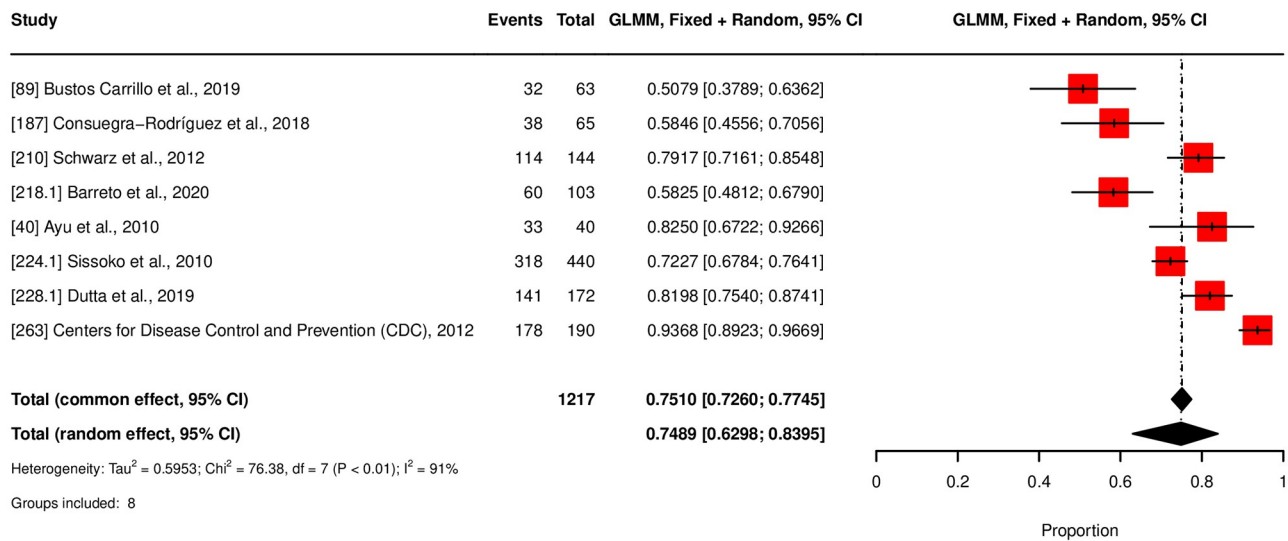

**Fig 5. Forest plot for symptomatic rate.**

population) is a realistic estimate for the general population. While this percentage is still relatively low compared to other arboviruses [29], mortality rates can be drastically higher in high-risk groups. Our meta-analysis revealed a mortality rate of 15.34% (15,340 per 100,000 population) in elderly and individuals with previous emergency department or intensive care admissions.

In defining the high-risk group for mortality, we included hospitalized patients who are typically older. As a result, the average age within this group was higher and advanced age is a recognized risk factor for increased mortality from CHIKV infection [30]. The task of separating the effects of comorbid conditions from the direct impact of CHIKV on mortality rates is complex. These factors often interact with each other, complicating the attribution of cause of death to CHIKV alone—particularly when our analysis could not conclusively establish the causes listed on death certificates. Furthermore, we recognize the possibility of publication bias in existing research on severe CHIKV cases. There may be an overrepresentation of studies focusing on severe outcomes and elevated mortality rates among individuals with underlying health complications or atypical presentations of CHIKV. Such a bias could lead to an overestimation of the mortality risk associated with the virus. Nonetheless, our SLR showed mortality rates up to 36.67% (36,670 per 100,000 population) in specific populations, demonstrating that despite its low rates in the general population, the impact of mortality should not be overlooked [27].

The chronic phase of chikungunya can be debilitating and long-lasting, leading to a significant health burden for individuals affected. Results from our meta-analysis showed a chronicity rate of 43.89% at three months, 34.39% at six months, and 31.87% at 12 months post-infection, indicating the lasting health burden. A meta-analysis conducted by Paixao and colleagues on the chronicity rate of chikungunya showed similar outcomes, with an overall no-recovery rate of 43% after three months [3]. One notable difference, possibly due to variations in methodologies, is that Paixao and colleagues reported slightly lower rates over time. Both studies indicate a stabilization over time, but more research is needed to comprehensively map the progression of the chronic phase. In conclusion, long-term chronic illness majorly impacts the quality of life of chikungunya patients [4, 6], making these results alarming, especially in light of the potential growing spread of the disease [9, 10].

The significant disease burden related to chikungunya was further underlined by an overall symptomatic rate of 74.9% in the general population. The symptomatic rate of chikungunya was estimated between 72% and 97% by the CDC Yellow Book, showing that our estimate could be on the low end [17]. A reason for this could be the various definitions of symptomatic manifestations across studies, which posed a challenge in deriving a precise estimate for this outcome. Additionally, estimates in the literature are mainly based on patients showing health-care-seeking behaviour, leaving out asymptomatic patients. Therefore, these estimates are likely to be overestimated. Because we created our estimate based on the total general population, we expect them to provide a better reflection of reality.

Two studies identified in the SLR were designed to investigate treatment options for Chikungunya and therefore included control groups. However, we excluded control populations without confirmed CHIKV from our analysis because our focus was on populations with confirmed CHIKV. In instances where multiple treatment options were assessed among confirmed CHIKV populations, these groups were included in the analysis as we aimed to understand the symptomatology of the disease at presentation in its acute phase. It should be noted that the inclusion of these populations did not significantly influence the outcomes of our study since the primary interest was in the manifestation of symptoms rather than treatment efficacy.

Although we obtained estimates for 11 of the 28 predefined endpoints, estimation for several endpoints proved infeasible due to their infrequent reporting as identified in the SLR. We did not obtain estimates for the underreporting factor, the length of the acute and the chronic phase, the duration of the different symptoms, and the frequencies of hospitalization and outpatient care. Even considering the subpopulation analysis method used, we could not estimate more endpoints. The limited number of studies reflects the uncertainty and novelty associated with chikungunya and the need for more research in this field.

In cases where meta-analyses were feasible for the endpoints, we encountered challenges due to poor data quality or absent data. This is attributable to two main reasons: firstly, the reporting of several endpoints varied inconsistently across studies, preventing their combination in a meta-analysis; and secondly, some studies that reported the desired endpoint did not meet the inclusion criteria, resulting in sparse data that hindered meaningful analysis. As a result, significant knowledge gaps persist regarding various aspects of chikungunya. Further research is necessary to fill these gaps and enhance our understanding of this disease. Additionally, consistent and strict reporting criteria on the clinical picture of chikungunya are needed to help create more comprehensive estimates. Enhanced quality and quantity of data could facilitate the possibility to study potential differences in symptomatology for the different CHIKV subtypes. Furthermore, it could enable investigations into the pathogenicity of CHIKV over the years by comparing data from previous outbreaks.

A strength of our study is the use of subpopulation analyses. We discovered that extracting subpopulations from individual studies allows more endpoints to be estimated, offering comparable populations that limit heterogeneity. The use of subgroups could be useful for future research and mitigate some of the data discrepancies detected in the SLR.

The main limitation of our study is the significant presence of heterogeneity indicated by an average $I^2$ statistic of 92%. This reflects substantial differences in the inclusion criteria among the studies, a tendency inherent in the disease area of CHIKV as shown by other meta-analyses reporting similar, or even higher, levels of heterogeneity [3]. There are several reasons for this high heterogeneity. First, data collection on chikungunya is conducted mostly during the outbreaks which limits the possibility of establishing strict scientific protocols as researchers must adapt to the dynamic nature of the event. Secondly, a standardized methodology for reporting endpoints is lacking, making it challenging to compare studies in a meta-analysis.

Thirdly, we noticed that including older individuals affected our results, by showing lower symptomatic rates but higher mortality and hospitalization rates. Future studies might exclude this demographic for more precise age-related outcomes. Additionally, other, less known, symptoms might have influenced the disease estimates. An example of this is depressive symptoms related to chikungunya. A study included in our analysis has potentially skewed our meta-analysis results with inflated estimates for fatigue, headache, and rash because they investigated depressive symptoms during the CHIKV infection [31]. This highlights how undisclosed factors that increase the population's vulnerability to chikungunya symptoms can potentially impact the research. Another limitation is the potential for confounding factors contributing to symptom prevalence, which we were unable control for in our study. There's an implicit assumption that the symptoms described have a causal association with Chikungunya; however, some symptoms such as myalgia and fatigue are commonly prevalent in the population and may not be causally related to CHIKV infection. The difficulty in establishing a direct causal relationship between these symptoms and CHIKV should be taken into consideration when interpreting the results. We acknowledge that this could affect the precision of the associations drawn in our analysis and suggest that future research should aim to discern the specific attributable risk of CHIKV for these symptoms. Lastly, outbreaks often occur in locations with limited surveillance systems, leading to lacking or less accurate data from these areas. The high heterogeneity shows the need for additional research in the fields, as well as standardized methodologies in studying chikungunya. Additionally, it emphasizes the importance of meta-analyses like these to come to accurate estimates.

## Conclusion

Chikungunya is recognized as a global public health threat, and the disease is expected to spread further due to globalization and climate change. At the same time, vector control and surveillance systems remain limited. Consequently, a thorough understanding of the clinical burden of chikungunya is important to inform public health intervention strategies and improve global surveillance. Our study showed that chikungunya poses a significant health burden, with 74.9% of infected individuals experiencing symptomatic disease. Chronic symptoms are found in 43.4% of patients and can be debilitating and long-lasting. We were not able to create pooled estimates on all endpoints, highlighting the still existing data gap in literature here. Nevertheless, the outcomes determined add to the growing body of evidence underlining the debilitating consequences of chikungunya. With the growing threat of chikungunya, health authorities and healthcare professionals must be prepared to adequately diagnose patients affected by the disease and consider public health interventions to limit its burden. Our findings contribute to the comprehension of chikungunya's clinical outcomes, essential for improving global surveillance and detecting potential outbreaks.

## Supporting information

**S1 Text. Literature search and study selection.** Containing the search strategy and PICOs of the studies included in the SLR.
(DOCX)

**S2 Text. Quality assessment tools.** Modified Downs & Black checklist and the NIH quality assessment tool.
(DOCX)

**S3 Text. PRISMA 2020 checklist.**
(DOCX)

**S1 Table. Quality assessment of included studies.**
(XLSX)

**S2 Table. Summary of study characteristics.**
(XLSX)

**S1 Fig. Forest plots of the clinical outcomes of chikungunya.**
(DOCX)

**S2 Fig. Influential case and outlier analysis with Baujat plots.**
(DOCX)

## Acknowledgments

We would like to thank the internal teams of Asc Academics who helped during the data extraction phase of the SLR, as well as Roma Kwiatkiewicz from Asc Academics for providing medical writing support.

## Author Contributions

**Conceptualization:** Adrianne M. de Roo, Timon Louwsma, Gerard T. Vondeling.

**Data curation:** Kris Rama, Timon Louwsma, Gabriel S. Gurgel do Amaral.

**Formal analysis:** Kris Rama.

**Investigation:** Hinko S. Hofstra, Gabriel S. Gurgel do Amaral.

**Methodology:** Kris Rama, Timon Louwsma.

**Project administration:** Timon Louwsma, Hinko S. Hofstra.

**Software:** Kris Rama.

**Supervision:** Gerard T. Vondeling, Maarten J. Postma, Roel D. Freriks.

**Visualization:** Kris Rama.

**Writing – original draft:** Kris Rama, Adrianne M. de Roo.

**Writing – review & editing:** Timon Louwsma.

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
