## [Decision Letter · Decision Letter 0]

9 Apr 2024

Dear Mr. Rama,

Thank you very much for submitting your manuscript "Clinical outcomes of chikungunya: A systematic literature review and meta-analysis" for consideration at PLOS Neglected Tropical Diseases. As with all papers reviewed by the journal, your manuscript was reviewed by members of the editorial board and by several independent reviewers. The reviewers appreciated the attention to an important topic. Based on the reviews, we are likely to accept this manuscript for publication, providing that you modify the manuscript according to the review recommendations. 

Your manuscript has been evaluated by 3 reviewers and all considered it a valuable contribution, but all also had comments for your consideration that may well improve your manuscript. Please consider these suggestions and modify your manuscript accordingly.

Sincerely,

Richard A. Bowen

Academic Editor

Mabel Carabali

Section Editor

Your manuscript has been evaluated by 3 reviewers and all considered it a valuable contribution, but all also had comments for your consideration that may well improve your manuscript. Please consider these suggestions and modify your manuscript accordingly.

Reviewer's Responses to Questions

**Key Review Criteria Required for Acceptance?**

**Methods**

-Are the objectives of the study clearly articulated with a clear testable hypothesis stated?

-Is the study design appropriate to address the stated objectives?

-Is the population clearly described and appropriate for the hypothesis being tested?

-Is the sample size sufficient to ensure adequate power to address the hypothesis being tested?

-Were correct statistical analysis used to support conclusions?

-Are there concerns about ethical or regulatory requirements being met?

Reviewer #1: The objectives were clearly state however a better description of the subpopulations within the text in methods and results would be helpful to add clarity to the study design. Although the results were only based on ~316 studies, it's not clear whether this number of studies and the sample size within these studies were adequately powered. I recommend that a statistician review the statistical methods in detail.

Reviewer #2: The methods are appropriate and the study has been well conducted and analysed.

Reviewer #3: S2 Checklist is incomplete. Please provide complete PRISMA checklist.

No other concerns about the Methods. The inclusion criteria are very broad, but this is acknowledged by the authors and probably necessary to cover the full range of symptomatology and clinical outcomes.

**Results**

-Does the analysis presented match the analysis plan?

-Are the results clearly and completely presented?

-Are the figures (Tables, Images) of sufficient quality for clarity?

Reviewer #1: The analysis matched the plan however as stated in the methods section, the subpopulations needed to be better described in the text of the manuscript. Page 5 lines 144-147 Please add the percent along with the N for each symptom. For example, rash constituted 171 studies of the 183 studies this would be 93%.

Page 5 line 166 The authors report mortality rate as a percentage. Please also calculate the mortality per 100,000 population. Many of the supplemental figures seem key to the authors conclusion yet they are hidden in supplemental figures. My suggestion is the figures most relevant to new findings should be pulled out of supplemental figures and added to the body of the manuscript. For example, adding the plots of those symptoms with the most variability between studies. Figures from the general population in S2 would be worthwhile adding to the main body of the manuscript. Figure 2 indicates a subpopulation of N=756 and then the flow chart bifurcates into exclusion based on newborns and children <15yr. It would be helpful to understand the number of studies excluded in each of these boxes.

Reviewer #2: There needs to be a fuller description of the studies included (apologies if I've missed this in supplementary materials). I appreciate this could become turgid, so care will have to be taken with writing, but readers need to understand a little more about the studies included to critically assess the results. An additional figure might clarify this. I would like to know the distinction between cohort and longitudinal studies, and what intersection there is between the cohort, longitudinal, prospective, and retrospective studies? How and why were symptoms studied in case-control studies, and what were the purposes of these studies. In addition, further information on specific groups of studies would help. Did the cohort studies (or some of them) follow a non-diseased group until (among some of them) disease developed, and then follow symptoms? I'd also like to know whether any of the studies included a control group.

Reviewer #3: Results are well presented in figures and tables. No further recommendations.

**Conclusions**

-Are the conclusions supported by the data presented?

-Are the limitations of analysis clearly described?

-Do the authors discuss how these data can be helpful to advance our understanding of the topic under study?

-Is public health relevance addressed?

Reviewer #1: The authors aimed to perform a comprehensive meta-analysis for chikungunya specifically focusing on specific symptoms, mortality and chronicity rates observed in both acute and convalescent phases of the illness. The manuscript provides important information however many of the findings are not unexpected. It would be interesting to compare studies with different subtypes (WA, ECSA, IOL, Asia and American subtypes) to determine if the subtypes have differences in symptoms, mortality or chronicity. Also, it would be interesting to compare outbreaks from several decades versus most recent outbreaks to determine if the virus has become more or less pathogenic over the years. Another concern was the extremely high mortality rate in the elderly of 15.4% and 36.67% in specific populations stated in the discussion section. This information does not seem likely considering that chikungunya has not historically caused mortality rates in these ranges even in the largest outbreaks in either Asia, Africa or the Americas. These numbers need to be better explained from where they were derived, the studies indicating these rates and their sample size. Also, whether the death certificates stated CHIKV as the cause of death. Many of the severe cases often are from people who have underlying conditions and the cause of death may not be specifically due to CHIKV. Finally, a better description of the subpopulations within the text in methods and results would be helpful to add clarity to the study design. Although the results were only based on ~316 studies, its not clear whether this number of studies and the sample size within these studies were adequately powered.

Reviewer #2: There's an implicit assumption that the symptoms described have a causal association to Chikungunya. Some symptoms such as myalgia and fatigue have quite high population prevalence and may not have a causal association to previously diagnosed CHIKV. I appreciate this can't be overcome in this study, but should be mentioned as a limitation.

Reviewer #3: Conclusions are not groundbreaking but further affirm most prevalent and frequent symptoms during CHIKV infection, and also provide additional data on chronicity of CHIKV infection and hospitalisation/ mortality rates.

**Editorial and Data Presentation Modifications?**

Reviewer #1: I suggested that some of the Figures should be placed in the main body of the manuscript instead of only in the supplemental sections.

Reviewer #2: At line 114 I think there is a typographical error; "commodities" should be "comorbidities". At line 141, substitute "one" for "less". It's better grammatically and more precise (assuming there weren't zero papers from some of these areas).

At line 264, what does "...challenges due to lack and/or data quality" mean? It needs to be rewritten. I wonder if the meaning is "poor quality or absent data"?

Reviewer #3: None - just to update table S2 and/ or include completed PRISMA diagram as best practice.

**Summary and General Comments**

Reviewer #1: See my comments in the conclusion section regarding the describing outbreaks over time and viral subtype analyses.

Reviewer #2: Please discuss critically your statement regarding the prevalence of arthralgia and arthritis (lines 181-2), including the definition for the latter. The definition of arthritis that I'm familiar with requires physician examination. Unless all studies included examination by a clinician this would be an important bias in estimation of arthritis prevalence. If none of the studies involved physician examination then arthritis could not be determined.

You state that you found 11 experimental studies (line 113). More information is needed on these. Were they the 11 studies on co-infection with Zika and Chikungunya? What was the study design? Were they clinical or epidemiological studies? If the latter, presumably randomised control trials?

Reviewer #3: Well-written study that provides additional evidence on the clinical endpoints of CHIKV.

Line 310: 's' missing on 'endpoint'.

PLOS authors have the option to publish the peer review history of their article (what does this mean?). If published, this will include your full peer review and any attached files.

Reviewer #1: No

Reviewer #2: Yes: David Harley

Reviewer #3: No

Figure Files:

Data Requirements:

Reproducibility:

References

---

## [Editor Report · Decision Letter 1]

28 May 2024

Dear Mr. Rama,

We are pleased to inform you that your manuscript 'Clinical outcomes of chikungunya: a systematic literature review and meta-analysis' has been provisionally accepted for publication in PLOS Neglected Tropical Diseases.

Best regards,

Richard A. Bowen

Academic Editor

Mabel Carabali

Section Editor

Thank you for the thoughtful responses to reviewer comments. I believe those changes have improved an already-valuable manuscript.